# Comparative Volume Analysis of Alveolar Defects by 3D Simulation

**DOI:** 10.3390/jcm8091401

**Published:** 2019-09-06

**Authors:** Pang-Yun Chou, Rafael Denadai, Rami R. Hallac, Sarayuth Dumrongwongsiri, Wei-Chuan Hsieh, Betty CJ Pai, Lun-Jou Lo

**Affiliations:** 1Department of Plastic and Reconstructive Surgery, Craniofacial Research Center, Chang Gung Memorial Hospital, Chang Gung University, Taoyuan City 33302, Taiwan; 2Analytical Imaging and Modeling Center, Department of Plastic Surgery, University of Texas Southwestern, Dallas, TX 75390, USA; 3Department of Plastic and Reconstructive Surgery, Chang Gung Memorial Hospital, Taoyuan City 33302, Taiwan; 4Division of Orthodontics, Department of Dentistry, Chang Gung Memorial Hospital, Taoyuan City 33302, Taiwan

**Keywords:** alveolar bone grafting, cleft, printed model, outcomes, 3D simulation, volume measurement

## Abstract

A precise volumetric assessment of maxillary alveolar defects in patients with cleft lip and palate can reduce donor site morbidity or allow accurate preparation of bone substitutes in future applications. However, there is a lack of agreement regarding the optimal volumetric technique to adopt. This study measured the alveolar bone defects by using two cone-beam computed tomography (CBCT)-based surgical simulation methods. Presurgical CBCT scans from 32 patients with unilateral or bilateral clefts undergoing alveolar bone graft surgery were analyzed. Two hands-on CBCT-based volumetric measurement methods were compared: the 3D real-scale printed model-based surgical method and the virtual surgical method. Different densities of CBCT were compared. Intra- and inter-examiner reliability was assessed. For patients with unilateral clefts, the average alveolar defect volumes were 1.09 ± 0.24 and 1.09 ± 0.25 mL (*p* > 0.05) for 3D printing- and virtual-based models, respectively; for patients with bilateral clefts, they were 2.05 ± 0.22 and 2.02 ± 0.27 mL (*p* > 0.05), respectively. Bland–Altman analysis revealed that the methods were equivalent for unilateral and bilateral alveolar cleft defect assessment. No significant differences or linear relationships were observed between adjacent different densities of CBCT for model production to obtain the measured volumes. Intra- and inter-examiner reliability was moderate to good (intraclass correlation coefficient (ICC) > 0.6) for all measurements. This study revealed that the volume of unilateral and bilateral alveolar cleft defects can be equally quantified by 3D-printed and virtual surgical simulation methods and provides alveolar defect-specific volumes which can serve as a reference for planning and execution of alveolar bone graft surgery.

## 1. Introduction

Secondary alveolar bone grafting (ABG) using autologous iliac crest bone tissue is a standard procedure for the management of patients with cleft lip and palate (CLP); a successful alveolar cleft defect repair produces maxillary arch continuity, provides adequate bony support, facilitates the eruption of permanent teeth, preserves periodontal health of teeth adjacent to the cleft, permits orthodontic tooth alignment, allows the placement of implants, and improves alar base symmetry [1,2].

Two-dimensional radiography has routinely been used for the diagnostic evaluation and treatment planning of maxillary alveolar cleft defect reconstructions [3,4]; however, this imaging modality is associated with drawbacks, such as no volumetric information, enlargement, distortion, and overlap of anatomical structures, and limitations for anatomical landmarks identification, thus affecting the accurate planning [5,6]. Conventional computed tomography (CT) scans have also been used to assess ABG-related outcomes because they provide precise and accurate representations of the anatomical structures and pathological processes [7,8], but CT is associated with high-dose ionizing radiation exposure, especially for patients at the developmental age [9,10]. Cone beam CT (CBCT) presents high-quality three-dimensional (3D) image acquisition and reconstruction parameters (including maxillary alveolar anatomical boundaries) and low radiation dose and cost. Consequently, CBCT has increasingly been adopted for ABG-related diagnostic and treatment assessments [7,8].

Despite these advances of imaging systems, the accuracy of determining the volume of maxillary alveolar cleft defect remains variable [7,8]. Defining an accurate volume of an alveolar defect is essential for ABG surgery because it helps the multidisciplinary cleft team to prepare for the procedure, such as in selecting the donor site and assessing the treatment outcome [7,8]. A sufficient quantity of cancellous bone grafting can be harvested from the anterior iliac crest [1,2,11]. However, this harvesting process involves the elevation of musculoperioteal flaps with significant dissection of bone and soft tissues, which leads to iatrogenic complications at the donor site (e.g., acute and chronic postoperative pain, paresthesia, seroma, hematoma, ambulation impairment, contour deformity, and scar-related cosmetic concern) and subsequent morbidity [1,2,11]. These morbidity features related to iliac crest bone graft harvesting may vary with the technique adopted and the quantity of bone harvested [12,13]. If the required volume for alveolar cleft defect reconstruction is defined preoperatively, a defect-specific quantity of iliac bone tissue can be harvested using minimally invasive techniques, thereby maximizing the unique properties of autologous tissue while minimizing the harvest volume, consequently reducing morbidity features related to complications at the donor site. This may improve the overall satisfaction with the treatment course by attenuating the burden of the longitudinal cleft rehabilitation process [14,15,16].

Recent systematic reviews have shown a lack of consensus regarding the adoption of different 3D imaging-based techniques for alveolar cleft defect assessments, and no gold standard method exists [7,8]. Despite the growing body of literature testing various CBCT-based methods for appraising alveolar cleft defects [7,8], no comparative study has analyzed the ABG-based surgical simulation tools using the 3D printed or virtual models. It is paramount that 3D surgical simulation models using CBCT imaging-based techniques be tested because they may have broad clinical and educational applications. This includes ABG planning and execution with the implementation of need-based iliac bone harvesting and grafting, implementing a shared decision-making process based on discussions between patients or parents and members of multidisciplinary cleft teams, and training residents and fellows (oral; ear, neck, and throat; head and neck; plastic; and maxillofacial surgeons). This may also serve as a benchmark for further investigations in dental, surgical, bioengineering, and nanotechnology disciplines [17,18], thereby improve future guidelines and recommendations for cleft ABG surgical care.

Accordingly, this study volumetrically measured alveolar cleft defects using two CBCT-based ABG surgical simulation models, namely 3D real-scale printed models versus 3D image virtual models. The authors hypothesized that both 3D surgical simulation models would present similar results for the volume parameter.

## 2. Material and Methods

Consecutive patients with non-syndromic unilateral or bilateral cleft lip and alveolus with or without palate who underwent CBCT scanning 2 weeks prior to ABG surgery between January and August 2018 were enrolled in the present study. All the included patients received primary surgeries and team management based on the current Chang Gung Craniofacial Center protocol, including lip repair at the age of 3 months, palatoplasty at the age of 9 months, and regular orthodontic visits before the ABG intervention [19,20]. To be qualified for ABG, patients were required to have acceptable dental alignment and adequate approximation of the alveolar segments. As indicated, some patients received presurgical regional orthodontic treatment or extraction of the interfering deciduous tooth. The patients who did not meet these criteria or who had any associated syndrome, or inadequate CBCT image data were excluded.

Ethical approval was obtained by the Institutional Review Board (201600968A3) and the study complied with the Declaration of Helsinki.

### 2.1. Alveolar Cleft Surgery Simulation Tools

To simulate alveolar cleft defect reconstruction and quantify the defect volume, two 3D hands-on methods were employed (Appendix A). For this, all CBCT scans were obtained using an i-CAT CBCT scanner (Imaging Sciences International, Hatfield, PA, USA) with the following parameters: 120 kVp, voxel size of 0.4 × 0.4 × 0.4 mm^3^, 40-s scan time, and 22 × 16-cm field of view. 3D volume renderings of the skull were generated by manual segmenting and adjusting the Hounsfield units (HU). Segmentation was performed twice for each patient using two random different densities (HU values) of CBCT within an acceptable image quality definition. For the first skull segmentation process, the HU value was manually adjusted until a detailed visualization of the anterior nasal spine, premaxilla region, and hard palatal shelves structures was achieved and with no dental braces-related artifacts. For the second skull segmentation process in the same patient, a different adjacent HU value was randomly adjusted using the first value as an initial reference point as well the same image quality definition. This process was performed by a trained health bioinformatics specialist (T.H.) who was a member of Chang Gung Craniofacial Imaging Laboratory.

To set the endpoint of simulated bone grafting using two different methods, the anatomical boundaries of alveolar defects were defined as follows: anterior and posterior borders were arch-aligned anteriorly and posteriorly, the superior border was aligned from anterior nasal spine up-tilt to lateral segment, and the inferior border was aligned along with bilateral cementoenamel junctions [21,22,23,24]. For the first simulation method (3D printing), two 3D real-scale models of maxillary alveolar defect were printed for each patient (Objet30 OrthoDesk 3D Printer, Stratasys Ltd., Israel) using a biocompatible PolyJet photopolymer material (MED610; Stratasys Ltd., Israel). Synthetic modeling clay (Canada Inc., Longueuil, QC, Canada) was adopted to reconstruct the alveolar defect (Figure 1). The volume (mL) of the modeling clay used was then measured using a water displacement technique [25,26].

For the second method (3D image virtual method), the ABG surgical simulation was performed using the SimPlant Pro software package (Materialize Dental, Leuven, Belgium). Initially, the 3D rendering of the skull was segmented at the level of the maxillary first premolar and mid-palatal regions (segmentation wizard tool). Using the bone point tool in axial, coronal, and axial planes, the alveolar defect was overfilled to virtually simulate the bone graft tissue. With the aid of Boolean operation and zoom tools, this simulated bone was sculpted using the anatomical boundaries of the defect as reference parameters. The volume (mL) of simulated bone grafted was finally calculated using the 3D object function (properties tool) (Figure 2 and Appendix A).

For both methods, all the simulations were performed twice by two independent board-certified cleft surgeons with a 2-week interval between each measurement session. The average value for each 3D printed and virtual model was adopted for analysis.

### 2.2. Statistical Analysis

In the descriptive analysis, data were presented as means ± standard deviations. Data distribution was verified using the Kolmogorov–Smirnov test, and paired *t*-test was adopted for comparisons. Bland–Altman plots [27] were obtained to assess the agreement of the 3D-printed model and image simulation methods. Box plots were generated to display the absolute difference in the two modified volumetric methods from the different densities of CBCT. Intraclass correlation coefficients (ICCs) were calculated for intra- and inter-examiner reliability for measurements using the two types of volumetric methods and densities of CBCT. A power analysis was performed prior to estimating an appropriate sample size based on the minimum ICC of 0.65, alpha (type I error) of 0.05 and power of 80% by two examiners/tools. Based on this analysis, at least 15 subjects were required for study. All tests were two-tailed, and *p* < 0.05 was considered statistically significant. All analyses were performed using SPSS Version 22.0 (IBM Corp., Armonk, NY, USA) and plotted using STATA 12.0 (StataCorp. 2011. Stata Statistical Software: Release 12. College Station, TX: StataCorp LP).

## 3. Results

A total of 128 CBCT-based 3D models (64 printed and 64 virtual models) from 32 patients with unilateral (n = 22, 13 boys and 9 girls, mean age 9.1 ± 0.2 years) or bilateral (n = 10, six boys and four girls, mean age 9.6 ± 0.7 years) clefts were used in this study.

### D-Based Alveolar Cleft Volume

For patients with unilateral cleft, the average alveolar defect volumes were 1.09 ± 0.24 and 1.09 ± 0.25 mL for 3D-printed and virtual simulation models, respectively; for patients with bilateral clefts, these values were 2.05 ± 0.22 and 2.02 ± 0.27 mL, respectively (Table 1). No significant differences were observed for the adopted densities of CBCT (Table 1).

Bland–Altman analysis revealed an agreement between the 3D-printed and virtual simulation models for unilateral and bilateral alveolar cleft-related measurements (Figure 3). Box plot interpretation showed no linear relationship between the two methods and the adopted densities of CBCT for unilateral and bilateral alveolar cleft-related measurements (Figure 4). Moderate and good intra- and inter-examiner reliability (all ICC > 0.6) were observed for all the measurements (Appendix A).

## 4. Discussion

In this study, volume-related comparisons between the two modified 3D imaging-guided measurement methods revealed no significant differences for unilateral and bilateral alveolar cleft defects. Preoperative CBCT-based estimation of the bone graft volume required for maxillary alveolar cleft reconstruction has been studied in recent years [7,8]. However, differences in sample compositions (age and type of CLP), measurement tools (landmark-, reference plane-, or mirror image-based methods, no definition of alveolar cleft boundaries, and no surgical simulation), and applied analysis (no comparisons between different methods or HU values or absence of reliability testing) preclude a comprehensive head-to-head comparison between the existing volume-related findings and the current results [7,8,21,28,29,30,31,32,33,34,35].

Studies using bone landmark-, reference plane-, or mirror image-guided methods (e.g., adopting the non-affected side or the straight-line axial, horizontal, and vertical reference planes to set the alveolar cleft region) have provided volumetric data for alveolar cleft defects [7,8,21,28,29,30,31,32,33,34,35]. However, the aforementioned studies did not apply all the potential information that can be retrieved from the CBCT imaging, such as the specific local anatomy of each patient and the opportunity for surgical simulation during volumetric-related data collection. In the current study, these methods were adapted by introducing the concept of ABG surgical simulation for measuring the alveolar defect-specific volume. By using the real-size printed and virtual models and defining anatomical boundaries as a target of grafting during simulation, trainees and surgeons can perform a simulated ABG surgery and retrieve a 3D-based prediction of the patient-specific anatomy with anticipation of the volume required for reconstruction and morphologic details such as shape and contour irregularities of the bone defect. Because the typically described pyramidal shape of alveolar cleft defect pattern is not necessarily the same among patients, these 3D imaging-guided methods also permit the preoperative definition of skeletal defect patterns through the establishment of the positional relationship between reference points (i.e., anterior nasal spine, lateral and medial alveolar cleft margins, and palatal shelf parameters), which is a key aspect for the planning and execution of ABG surgery. Moreover, because trainees and surgeons can manipulate the printed or virtual models to explore the details of anatomical boundaries, the adopted techniques demonstrate advantages with respect to estimating the volume in the operating room (restricted mobilization of the patients’ head and limited visual field due to soft tissue restrictions even after wide dissections and release of the periosteum and scarring tissues). Therefore, it was expected that the actual need for bone graft tissue could be underestimated intraoperatively compared with the tested 3D simulation methods. Underestimation or overestimation during surgery may increase the operative time (i.e., by requiring the harvesting of more iliac bone) or donor site morbidity, respectively.

Different densities of CBCT (HU values) have been adopted to improve the imaging quality [36,37], but the adopted HU values have no effect on or relationship with volume measurements in bilateral and unilateral clefts, indicating that the interference from human error during the choice of HU when segmenting the bony boundaries was acceptable and that the most representative anatomical regions could be well identified. Though manual adjustment for the selected HU is relatively subjective, the serial 50 HU gap could be exactly determined based on clear identification of the anatomical boundaries in an experienced hand. The moderate-to-good intra- and inter-rater reliability values for all the tested parameters demonstrated consistency among the collected volumetric data. These CBCT-based methods thus provided a method for accurate volumetric quantification of the maxillary alveolar defect in patient with unilateral and bilateral cleft. As errors have been described for HU derived from CBCT systems, future investigation should test whether different technical modalities, e.g., CBCT-derived HU using linear attenuation coefficients [38], present (or not) interference with the volume measurements of alveolar bone defects. 

For centers with 3D imaging facilities, 3D-guided planning and execution of ABG surgery is advocated because it can provide patient-specific information, including need-based volume and valuable details of anatomical features. Although both models provided adequate measurement for the alveolar defect volumes, certain manipulation-related issues should be considered. In both methods, the 3D models can be rotated as needed to accurately place the simulated bone material exactly at the defining borders of the alveolar defect. The virtual method can provide reliable and accurate defect-specific information for orthodontists and surgeons with no need for 3D printed models, thus reducing the overall cost of ABG surgical planning. By contrast, the 3D printed model provides an improved surgical scenario because it mimics a real operation environment by employing the manipulation of surgical instruments, the representation of alveolar defects, and modeling clay (simulating bone graft tissue). Each center should therefore judge the applicability of each tool when introducing a specific method in their clinical practice and educational programs.

This study is not without limitations. The current sample restricted patients according to their age at the time of undergoing secondary ABG procedures, with extrapolations for primary or tertiary ABG procedures requiring careful interpretations. The difference in alveolar cleft volume between patients with and without presurgical regional orthodontic treatment were not assessed, which is a topic deserving of future study with a larger sample size. The absence of simulated soft tissues (nasal floor, palatal, and gingival tissues) does not limit simulated ABG surgeries because the examiners can easily identify the extent of the defect, including the confluence of the structures in the posterior region, which constitutes a challenging area in actual surgery. Further studies may consider the soft tissue perspective when using the described models. Additional investigation is also required to provide a cost-effectiveness analysis for facilitating the decision-making process of ABG care. Education-based research involving numerous surgeons with various levels of expertise should assess the impact of the described methods with financial, logistic, and time constraints in teaching cleft ABG surgery. Structured surveys may also reveal the preferences and perceptions of trainees and surgeons when performing ABG procedures using the 3D printed and virtual models. Although it was demonstrated that both 3D printed and virtual models may be interchangeably used for the measurement of alveolar cleft volume, a clinically controlled trial is required to test the present findings according to the clinical reality. For this, it is reasonable to apply the current unilateral and bilateral cleft defect-specific volumetric data for future ABG surgery-based research, with anatomy-related aspects being evaluated intraoperatively on a case-by-case basis. This may potentially reduce the operative time and donor site-related morbidity and thus also merit future investigation. 

The appraisal of this volumetric data may also provide useful information for clinical practice. It was demonstrated that defect volumes of approximately 1.0 mL and 2.0 mL were estimated to be present in unilateral and bilateral alveolar cleft defect reconstructions, respectively. This agrees with the findings of most relevant studies [7,8,21,28,29,30,31,32,33,34,35]. To transfer these 3D imaging-based data to actual surgery, the desired harvested bone grafts should be based on the upper limits (1.5 mL and 2.5 mL for unilateral and bilateral cleft lip and alveolus with or without palate, as displayed in Figure 3), instead of the average of cleft defect volumes, because most of the cleft defects will be fitted within this value. Because of the availability and costs (health insurance or government-based restrictions) related to 3D imaging models, the findings of this study are particularly useful for professionals treating CLP in developing countries, in which most of the children born with CLP worldwide reside [14,39,40].

## 5. Conclusions

This study reveals that the volume of unilateral and bilateral alveolar cleft defects can be quantified with equal performance levels by 3D-printed and virtual-based surgical simulation methods and provides alveolar defect-specific volumes which may be adopted as a reference point for the planning and execution of ABG surgery.

## Figures and Tables

**Figure 1 jcm-08-01401-f001:**
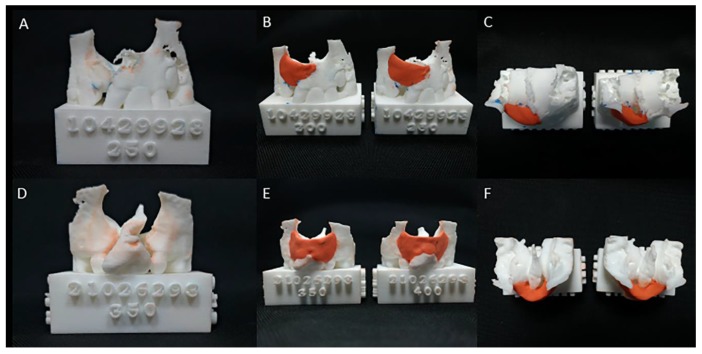
The 3D-printed models for alveolar bone graft surgery simulation. (**A**) Unilateral alveolar cleft models displaying (**B**) the superior, inferior, (**C**) anterior, and posterior borders of unilateral alveolar cleft defects filled with simulated bone graft tissue (red modeling clay) in two models with different densities of computed tomography (CBCT) from the same patient with complete unilateral cleft lip and palate. (**D**) Bilateral alveolar cleft models displaying (**E**) the superior, inferior, (**F**) anterior, and posterior borders of a bilateral cleft defect filled with simulated bone graft tissue (red modeling clay) in two models with different densities of CBCT from the same patient with complete bilateral cleft lip and palate.

**Figure 2 jcm-08-01401-f002:**
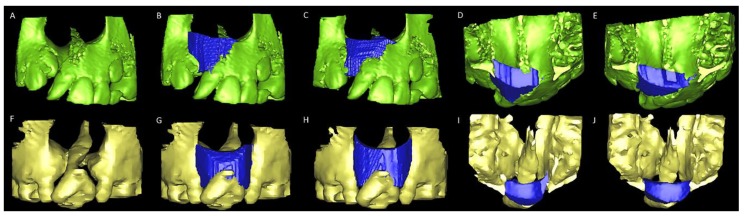
Virtual models for alveolar bone graft surgery simulation. (**A**) Unilateral alveolar cleft models displaying (**B**,**C**) the superior, inferior, (**D**,**E**) anterior, and posterior borders of bone defects filled with simulated bone graft tissue (blue) in two models with different densities of CBCT from the same patient with complete unilateral. (**F**) Bilateral alveolar cleft models displaying (**G**,**H**) the superior, inferior, (**I**,**J**) anterior, and posterior borders of bone defects filled with simulated bone graft tissue (blue) in two models with different densities of CBCT from the same patient with complete bilateral cleft lip and palate.

**Figure 3 jcm-08-01401-f003:**
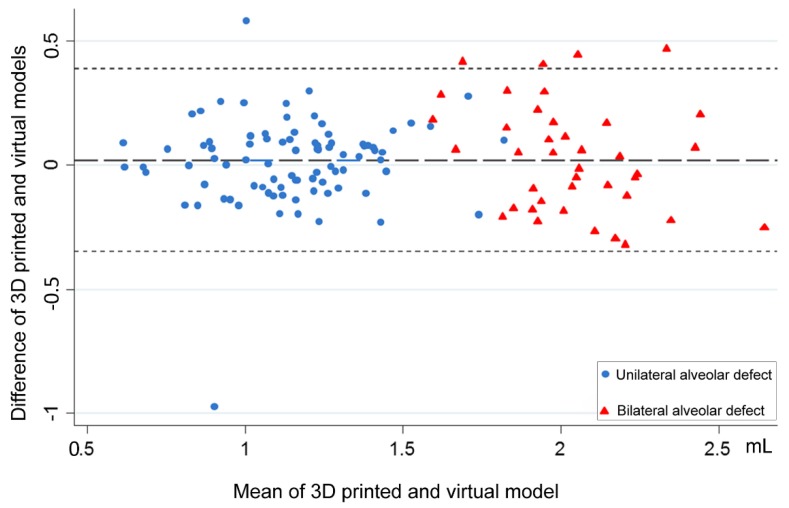
Bland–Altman plot indicating that the mean values of the difference in the volume of the alveolar defect between 3D-printed and virtual-based surgical simulation methods are approximately at the zero-line and are equally distributed within the zero-line, revealing that the two methods produced similar results.

**Figure 4 jcm-08-01401-f004:**
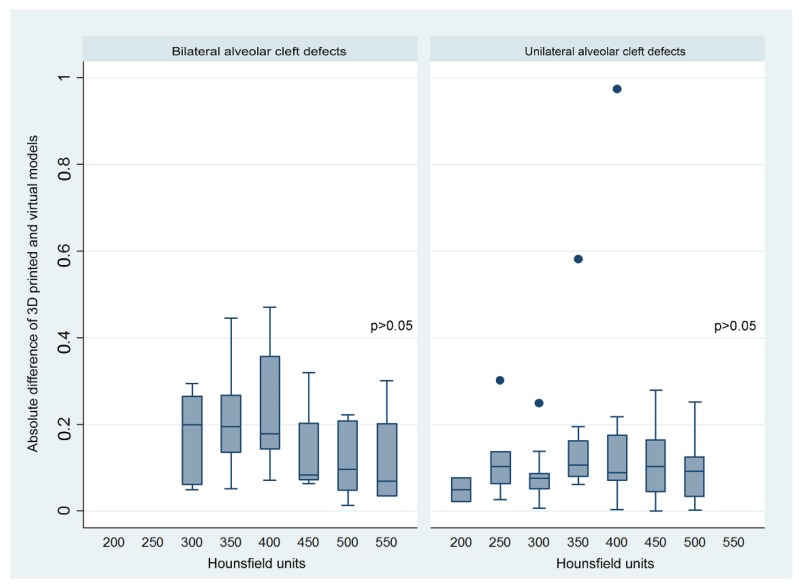
The absolute difference between 3D-printed and virtual-based surgical simulation methods showed no linear relationship with different Hounsfield units adopted for bilateral and unilateral alveolar cleft defects.

**Table 1 jcm-08-01401-t001:** 3D imaging-based volume of unilateral and bilateral alveolar cleft defects by printed and virtual models.

Parameters	3D Printed Model	Virtual Model	*p*-Value
**Unilateral alveolar cleft volume**, mL			
HU1 400 (p25 = 300, p75 = 450)	1.07 ± 0.25	1.05 ± 0.27	>0.05
HU2 375 (p25 = 350, p75 = 462.5)	1.11 ± 0.23	1.14 ± 0.23	>0.05
Total	1.09 ± 0.24	1.09 ± 0.25	>0.05
**Bilateral alveolar cleft volume**, mL			
HU1 350 (p25 = 300, p75 = 350)	2.07 ± 0.24	2.03 ± 0.29	>0.05
HU2 400 (p25 = 350, p75 = 400)	2.02 ± 0.20	2.00 ± 0.25	>0.05
Total	2.05 ± 0.22	2.02 ± 0.27	>0.05

Data presented as mean ± standard deviation; HU1, first Hounsfield unit; HU2, second Hounsfield unit; p25, 25th percentile; p75, 75th percentile.

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
