# Peer review of "Comparative Volume Analysis of Alveolar Defects by 3D Simulation"

_jcm, 2019, doi:10.3390/jcm8091401_

Round 1
Reviewer 1 Report
Comments and suggestions for authors
This manuscript might be worthy of publication due to presenting a topic that needs to be further elucidated.
However, as presented, this manuscript needs some major modifications to ease the understanding and to be suitable for publication. Generally, the language needs to be improved. Despite some grammatical and spelling errors several examples of too long sentences are present and some sentences are presented in an awkward manner. E.g. in rows 19-23, 55-59, 64-69, 80-82.
Furthermore, presenting a comparison between two measurement methods, it would be more interesting and appropriate if the study compared the actual clinical outcome, e.g. comparing your findings with actual grafted bone volume. Or comparing the two volume analyzing methods on patient postoperative morbidity. Perhaps also adding a control group were no preoperative measurements were made?
I comment the manuscript in the sub sectional order as follows:
Title
Appropriate, informative and short. Does not need to be corrected.
Abstract
In row 19 withdraw “A single”.
The statement in row 20, that the donor site morbidity is reduced due to a less quantity of harvested bone volume, is not supported in the references. This statement is crucial for the project objective and must be supported by previous scientific publication. Perhaps the manuscript is presentable without this statement but if you want to keep it I suggest appropriate references. Later in the manuscript the same statement appears I row 69-70. The references mentioned is References 21 and 22. Neither of them is claiming that donor site morbidity is correlated to the harvested volume.
The result presented in row 30-31 (and later on in row 172-173) is not a result from this study. A higher volume defect in bilateral clefts than in unilateral - This we already knew. Perhaps you can keep this as a finding that supports the accuracy of the measurement technique, but as a result; suggest withdrawal.
Introduction
The introduction is clear and appropriate. Despite the issues in references and language, no further corrections are needed. The aim has to be clearer. What are your hypotheses? Why do you want to compare these methods and for what means?
Methods
Generally appropriate and clear. However, the supplementary data are missing in this preview such as Figure S1 (late in results even Table S1). Further, even if dual adjustments of HU was made, I am sceptic to the manually adjustment and that no standard method or standard value of HU was used. You present a result with minor differences and, to my experience, I suspect that the manually adjustment (even if made twice) is at great risk of having the results skewed. At least you need to comment on this in the discussion.
In the methods section you present no real analysis of the reliability test. In row 153-154 you state that a mean value is used of the two measured occasions. This is just a mean value and do not represent a true reliability test. Later in the statistical analysis section you are mention ICC. You need to more thoroughly and separately describe both the intra-individual and the inter-individual reliability test in the methods section.
Results
As mentioned earlier, the result presented in row 172-173 is of minor interest and the supplementary data are missing in this preview such as Table S1.
Discussion
This is the part of the manuscript where we suggest a major revision. We suggest you start discussion the results and later the method. The discussion should be reference based on comparison with previous findings. Further limitations are discussed and lastly you suggest implementation of your results.
In the present manuscript row 193-209 is no discussion but repeated introduction; suggests withdrawal. Row 210-227 is a continuation of this repeated introduction mixed with suggestion of clinical implementation; suggests withdrawal of the majority and rearrangement.
In row 228 the discussion starts but further corrections are needed:
Row 235-237
This is not shown in the present study. The amount of bone graft is not shown. For this you need to compare the findings with the actual clinical harvesting. What you did was to compare two measurement methods with each other without comparing with the actual bone volume used.
Row 251-252
The same as above. You did not show what the actual bone volume was. You did compare two measurement methods with each other without comparing with the actual bone volume used.
Row 252-254
This is an awkward sentence. You cannot start claiming that you did find conformity with several relevant studies while you are finishing the same sentence claiming that you did not find conformity. Either you did find conformity or not.
Row 260-264
This is not enough evidence in your study to implement this clinically; suggest you write this sentence as suggestions for future studies.
Row 269
Here you state a need for future studies. I suggest that you first of all mention a clinically controlled trial to test the present findings according to the clinical reality before you look further into CEA or decision making. This is also applicable in row 275-276.
Conclusion
Appropriate, clear and well supported in the results. No corrections needed
References
Appropriate number and date of the references. However, almost all references are mentioned in the introduction and repeatedly used in the discussion. I suggest you put in new references in the discussion and decrease the number of references in the introduction. E.g. you do not need 11 references to exemplify previous volume measurements of maxillary alveolar cleft defects (row 61).
References 21 and 22 is aiming to support the statement of minor morbidity as a result of less harvested bone quantity (row 20 and 69-70). Neither of them is claiming that donor site morbidity is correlated to the harvested volume. Please correct.
Author Response
Reviewer 1
This manuscript might be worthy of publication due to presenting a topic that needs to be further elucidated.
Response: We would like to express our gratitude to reviewer, as he/she comments have improved our manuscript.
However, as presented, this manuscript needs some major modifications to ease the understanding and to be suitable for publication. Generally, the language needs to be improved. Despite some grammatical and spelling errors several examples of too long sentences are present and some sentences are presented in an awkward manner. E.g. in rows 19-23, 55-59, 64-69, 80-82.
Response: The manuscript was reviewed by an English editing company as requested. The sentences in rows 19-23, 55-59, 64-69, 80-82 were modified as requested.
Furthermore, presenting a comparison between two measurement methods, it would be more interesting and appropriate if the study compared the actual clinical outcome, e.g. comparing your findings with actual grafted bone volume. Or comparing the two volume analyzing methods on patient postoperative morbidity. Perhaps also adding a control group were no preoperative measurements were made?
Response: This study volumetrically measured alveolar cleft defects by using two CBCT-based ABG surgical simulation models. The lack of further comparative groups as well as morbidity-related data were addressed as limitations of current study, deserving future investigations.
Abstract In row 19 withdraw “A single”.
Response: This was reviewed as requested.
The statement in row 20, that the donor site morbidity is reduced due to a less quantity of harvested bone volume, is not supported in the references. This statement is crucial for the project objective and must be supported by previous scientific publication. Perhaps the manuscript is presentable without this statement but if you want to keep it I suggest appropriate references. Later in the manuscript the same statement appears I row 69-70. The references mentioned is References 21 and 22. Neither of them is claiming that donor site morbidity is correlated to the harvested volume.
Response: We replaced the two references by other two references. The order and number of references were also adjusted as requested. Now the adopted references (numbers 12 and 13) have formally addressed the issue of harvested volume:
Loeffler BJ, Kellam JF, Sims SH, Bosse MJ. Prospective observational study of donor-site morbidity following anterior iliac crest bone-grafting in orthopaedic trauma reconstruction patients. J Bone Joint Surg Am. 2012 Sep 19;94(18):1649-54.
“Donor-site morbidity may vary depending on the harvest site, technique, and volume of bone graft harvested”
Canady JW, Zeitler DP, Thompson SA, Nicholas CD. Suitability of the iliac crest as a site for harvest of autogenous bone grafts. Cleft Palate Craniofac J. 1993 Nov;30(6):579-81.
“The amount of bone harvested, the age of patient, and the surgical technique used obviously have a definite impact on the complications expected or encountered.”
The result presented in row 30-31 (and later on in row 172-173) is not a result from this study. A higher volume defect in bilateral clefts than in unilateral - This we already knew. Perhaps you can keep this as a finding that supports the accuracy of the measurement technique, but as a result; suggest withdrawal.
Response: The sentence was excluded from body of abstract and results section as requested.
Introduction The introduction is clear and appropriate. Despite the issues in references and language, no further corrections are needed. The aim has to be clearer. What are your hypotheses? Why do you want to compare these methods and for what means?
Response: The issues in references and language were addressed as requested. We introduced the hypotheses and why we do you want to compare these methods and for what means.
Methods Generally appropriate and clear. However, the supplementary data are missing in this preview such as Figure S1 (late in results even Table S1). Further, even if dual adjustments of HU was made, I am sceptic to the manually adjustment and that no standard method or standard value of HU was used. You present a result with minor differences and, to my experience, I suspect that the manually adjustment (even if made twice) is at great risk of having the results skewed. At least you need to comment on this in the discussion.
Response: The supplementary files were provided as requested. The adjustment of HU was addressed in discussion section as requested.
In the methods section you present no real analysis of the reliability test. In row 153-154 you state that a mean value is used of the two measured occasions. This is just a mean value and do not represent a true reliability test. Later in the statistical analysis section you are mention ICC. You need to more thoroughly and separately describe both the intra-individual and the inter-individual reliability test in the methods section.
Response: The reliability test issue was addressed in Methods section as requested. For both 3D methods, all simulations were performed twice by two independent board-certified cleft surgeons with a 2-week interval between each measurement session. The average value for each 3D printed and virtual model was adopted for analysis, whereas all values were considered for reliability testing. The supplementary Table 1 displays that moderate and good intra- and inter-examiner reliability (all ICC > 0.6) were observed for all measurements.
Results As mentioned earlier, the result presented in row 172-173 is of minor interest and the supplementary data are missing in this preview such as Table S1.
Response: The result presented in row 172-173 was deleted as requested. The supplementary files were provided as requested.
Discussion This is the part of the manuscript where we suggest a major revision. We suggest you start discussion the results and later the method. The discussion should be reference based on comparison with previous findings. Further limitations are discussed and lastly you suggest implementation of your results.
Response: The order of discussion was modified as requested. Results first, then method, comparisons with previous findings, limitations and implementation of the results are sequenced accordingly.
In the present manuscript row 193-209 is no discussion but repeated introduction; suggests withdrawal. Row 210-227 is a continuation of this repeated introduction mixed with suggestion of clinical implementation; suggests withdrawal of the majority and rearrangement.
Response: The text was rearranged as requested.
In row 228 the discussion starts but further corrections are needed:
Response: The text was rearranged as requested.
Row 235-237. This is not shown in the present study. The amount of bone graft is not shown. For this you need to compare the findings with the actual clinical harvesting. What you did was to compare two measurement methods with each other without comparing with the actual bone volume used.
Response: The sentence was reformulated as requested.
Row 251-252. The same as above. You did not show what the actual bone volume was. You did compare two measurement methods with each other without comparing with the actual bone volume used.
Response: The sentence was reformulated as requested.
Row 252-254. This is an awkward sentence. You cannot start claiming that you did find conformity with several relevant studies while you are finishing the same sentence claiming that you did not find conformity. Either you did find conformity or not.
Response: The sentence was reformulated as requested.
Row 260-264. This is not enough evidence in your study to implement this clinically; suggest you write this sentence as suggestions for future studies.
Response: The sentence was reformulated as requested.
Row 269. Here you state a need for future studies. I suggest that you first of all mention a clinically controlled trial to test the present findings according to the clinical reality before you look further into CEA or decision making. This is also applicable in row 275-276.
Response: The order of sentences was readjusted as requested.
References Appropriate number and date of the references. However, almost all references are mentioned in the introduction and repeatedly used in the discussion. I suggest you put in new references in the discussion and decrease the number of references in the introduction. E.g. you do not need 11 references to exemplify previous volume measurements of maxillary alveolar cleft defects (row 61).
Response: The references in the introduction and methods sections were reduced and increased, respectively.
References 21 and 22 is aiming to support the statement of minor morbidity as a result of less harvested bone quantity (row 20 and 69-70). Neither of them is claiming that donor site morbidity is correlated to the harvested volume. Please correct.
Response: The references 21 and 22 were modified as requested.
Reviewer 2 Report
This study used two CBCT-based surgical simulation methods, 3D printing and virtual simulation, to measure the volume of alveolar defects among patients with cleft lip and palate. The results showed that these two methods can equally quantify the volume of alveolar defects, which can serve as a reference for subsequent surgical planning. This study is clinically significant, interesting and in general well designed. However, the sample size seems to be small while the sample size estimation and power analysis remain unclear. It would be also be helpful if more information about the time and cost of two methods can be included in the discussion session. This manuscript is in general well written. However, the first sentence of the Abstract says: "A single A precise volumetric assessment", where "A single A precise" seems to be a typo or an error caused by formatting.
Author Response
Reviewer 2
This study used two CBCT-based surgical simulation methods, 3D printing and virtual simulation, to measure the volume of alveolar defects among patients with cleft lip and palate. The results showed that these two methods can equally quantify the volume of alveolar defects, which can serve as a reference for subsequent surgical planning. This study is clinically significant, interesting and in general well designed. However, the sample size seems to be small while the sample size estimation and power analysis remain unclear. It would be also be helpful if more information about the time and cost of two methods can be included in the discussion session. This manuscript is in general well written. However, the first sentence of the Abstract says: "A single A precise volumetric assessment", where "A single A precise" seems to be a typo or an error caused by formatting.Response: We would like to express our gratitude to reviewer, as he/she comments have improved our manuscript.
A power analysis was performed prior to estimating an appropriate sample size based on the minimum intra-class correlation coefficient of 0.65, alpha (type I error) of 0.05 and power of 0.8 by two raters/tools. Based on this analysis, at least 15 subjects would be required for study. This was added to statistical analysis subhead as requested.
For this study, we do not collect the time and cost for the tested methods. This was described as a limitation of study in discussion section, deserving further investigations.
The manuscript was reviewed by MDPI English editing as requested.